# COALA: Co-Aligned Autoencoders for Learning Semantically Enriched Audio Representations

**Xavier Favory** [* 1]  **Konstantinos Drossos** [* 2]  **Tuomas Virtanen** [2]  **Xavier Serra** [1]

## Abstract

Audio representation learning based on deep neural networks (DNNs) emerged as an alternative approach to hand-crafted features. For achieving high performance, DNNs often need a large amount of annotated data which can be difficult and costly to obtain. In this paper, we propose a method for learning audio representations, aligning the learned latent representations of audio and associated tags. Aligning is done by maximizing the agreement of the latent representations of audio and tags, using a contrastive loss. The result is an audio embedding model which reflects acoustic and semantic characteristics of sounds. We evaluate the quality of our embedding model, measuring its performance as a feature extractor on three different tasks (namely, sound event recognition, and music genre and musical instrument classification), and investigate what type of characteristics the model captures. Our results are promising, sometimes in par with the state-of-the-art in the considered tasks and the embeddings produced with our method are well correlated with some acoustic descriptors.

## 1. Introduction

Legacy audio-based machine learning models were trained using sets of handcrafted features, carefully designed by relying on psychoacoustics and signal processing expert knowledge. Recent approaches are based on learning such features directly from the data, usually by employing deep learning (DL) models (Bengio et al., 2013; Hershey et al., 2017; Pons et al., 2017a), often making use of manually annotated datasets that are tied to specific applications (Tzanetakis & Cook, 2002; Marchand & Peeters, 2016; Salamon et al., 2014). Achieving high performance with DL-based

---
[*]Equal contribution  [1]Music Technology Group, Universitat Pompeu Fabra, Barcelona, Spain [2]Audio Research Group, Tampere University, Tampere, Finland. Correspondence to: Xavier Favory <xavier.favory@upf.edu>.

*Published at the workshop on Self-supervision in Audio and Speech at the $37^{th}$ International Conference on Machine Learning*, Vienna, Austria. Copyright 2020 by the author(s).

methods and models, often requires sufficient labeled data which can be difficult and costly to obtain, especially for audio signals (Favory et al., 2018). As a way to lift the restrictions imposed by the limited amount of audio data, different published works employ transfer learning on tasks were only small datasets are available (Yosinski et al., 2014; Choi et al., 2017). Usually in such a scenario, an embedding model is first optimized on a supervised task for which a large amount of data is available. Then, this embedding model is used as a pre-trained feature extractor, to extract input features that are used to optimize another model on a different task, where a limited amount of data is available (Van Den Oord et al., 2014; Choi et al., 2017; Pons & Serra, 2019a; Alonso-Jiménez et al., 2020).

Recent approaches adopt self-supervised learning, aiming to learn audio representations on a large set of unlabeled multimedia data, e.g. by exploiting audio and visual correspondences (Aytar et al., 2016; Arandjelovic & Zisserman, 2017). Such approaches have the advantage of not requiring manual labelling of large amount of data, and have been successful for learning audio features that can be used in training simple, but competitive classifiers (Cramer et al., 2019). Different approaches focus on learning audio representations by employing a task-specific distance metric and weakly annotated data. For example, the triplet-loss can be used to maximize the agreement between different songs of same artist (Park et al., 2017) or a contrastive loss can enable maximizing the similarity of different transformations of the same example (Chen et al., 2020). Other approaches leverage images and their associated tags to learn content-based representations by aligning autoencoders (Schonfeld et al., 2019). However the alignment is done by optimizing cross-reconstruction objectives, which can be overly complex for learning data representations.

In our work we are interested in learning audio representations that can be used for developing general machine listening systems, rather than being tied to a specific audio domain. We take advantage of the massive amount of online audio recordings and their accompanying tag metadata, and learn acoustically and semantically meaningful features. To do so, we propose a new approach inspired from image and the natural language processing fields (Schonfeld et al., 2019; Silberer & Lapata, 2014), but we relax the alignment

objective by employing a contrastive loss (Chen et al., 2020), in order to co-regularize the latent representations of two autoencoders, each one learned on a different modality.

The contributions of our work are:

- We adapt a recently introduced constrastive loss framework (Chen et al., 2020), and we apply it for audio representation learning in a heterogeneous setting (the embedding models process different modalities).
- We propose a learning algorithm, combining a contrastive loss and an autoencoder architecture, for obtaining aligned audio and tag latent representations, in order to learn audio features that reflect both semantic and acoustic characteristics.
- We provide a thorough investigation of the performance of the approach, by employing three different classification tasks.
- Finally we conduct a correlation analysis of our embeddings with acoustic features in order to get more understanding of what characteristics they capture.

The rest of the paper is as follows. In Section 2 we thoroughly present our proposed method. Section 3 describes the utilized dataset, the tasks and metrics that we employed for the assessment of the performance, the baselines that we compare our method with, and the correlation analysis with acoustic features that we conducted. The results of these evaluation processes are presented and discussed in Section 4. Finally, Section 5 concludes the paper and proposes future research directions.

## 2. Proposed method

Our method employs two different autoencoders (AEs) and a dataset of multi-labeled annotated (i.e. multiple labels/tags per example) time-frequency (TF) representations of audio signals, $\mathbb{G} = \{(\mathbf{X}_a^q, \mathbf{y}_t^q)\}_{q=1}^Q$, where $\mathbf{X}_a^q \in \mathbb{R}^{N \times F}$ is the TF representation of audio, consisting of $N$ feature vectors with $F$ log mel-band energies, $\mathbf{y}_t^q \in \{0,1\}^C$ is the multi-hot encoding of tags for $\mathbf{X}_a^q$, out of a total of $C$ different tags, and $Q$ is the amount of paired examples in our dataset. These tags characterize the content of each corresponding audio signal (e.g. "kick", "techno", "hard").

The audio TF representation and the associated multi-hot encoded tags of the audio signal, are used as inputs to two different AEs, one targeting to learn low-level acoustic features for audio and the other learning semantic features (for the tags), by employing a bottleneck layer and a reconstruction objective. At the same time, the learned low-level features of the audio signal are aligned with the learned semantic features of the tags, using a contrastive loss. All employed modules are jointly optimized, yielding an audio encoder that provides audio embeddings capturing both low-level acoustic characteristics and semantic information regarding the contents of the audio. An illustration of our

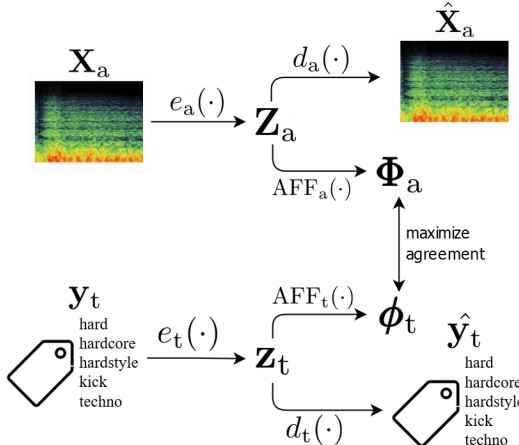

*Figure 1.* Illustration of our proposed method. $\mathbf{Z}_a$ and $\mathbf{z}_t$ are aligned through maximizing their agreement and, at the same time, are used for reconstructing back the original inputs.

method is in Figure 1.

### 2.1. Learning low-level audio and semantic features

For learning low-level acoustic features from the input audio TF representation, $\mathbf{X}_a$[1], we employ a typical AE structure based on convolutional neural networks (CNNs) and on having a reconstruction objective. Since AEs have proven to be effective in unsupervised learning of low-level features in different tasks and especially in audio (Van Den Oord et al., 2017; Amiriparian et al., 2017; Mimilakis et al., 2018; Drossos et al., 2018), our choice of the AE structure followed naturally.

The AE that processes $\mathbf{X}_a$ is composed of an encoder $e_a(\cdot)$ and a decoder $d_a(\cdot)$, parameterized by $\theta_{ea}$ and $\theta_{da}$ respectively. $e_a$ accepts $\mathbf{X}_a$ as an input and yields the learned latent audio representation, $\mathbf{Z}_a \in \mathbb{R}_{\geq 0}^{K \times T' \times F'}$. Then, $d_a$ gets $\mathbf{Z}_a$ as input and outputs a reconstructed version of $\mathbf{X}_a$, $\hat{\mathbf{X}}_a$, as

$$\mathbf{Z}_a = e_a(\mathbf{X}_a; \theta_{ea}), \text{ and} \tag{1}$$

$$\hat{\mathbf{X}}_a = d_a(\mathbf{Z}_a; \theta_{da}). \tag{2}$$

We model $e_a$ using a series of convolutional blocks, where each convolutional block consists of a CNN, a normalization process, and a non-linearity. As a normalization process we employ the batch normalization (BN), and as a non-linearity we employ the rectified linear unit (ReLU). The process for each convolutional block is

$$\mathbf{H}^{l_e} = \text{ReLU}(\text{BN}^{l_e}(\text{CNN}^{l_e}(\mathbf{H}^{l_e-1}))), \tag{3}$$

where $l_{ea} = 1, \ldots, N_{\text{CNN}}$ is the index of the convolutional block, $\mathbf{H}^{l_{ea}} \in \mathbb{R}_{\geq 0}^{K_{l_{ea}} \times T'_{l_{ea}} \times F'_{l_{ea}}}$ is the $K_{l_{ea}}$ learned feature maps of the $l_{ea}$-th CNN, $\mathbf{H}^{N_{\text{CNN}}} = \mathbf{Z}_a$, and $\mathbf{H}^0 = \mathbf{X}_a$.

---

[1] For the clarity of notation, the superscript $q$ is dropped here and for the rest of the document, unless it is explicitly needed.

Audio decoder, $d_a$, is also based on CNNs, but it employs transposed convolutions (Radford et al., 2016; Dumoulin & Visin, 2016) in order to expand $\mathbf{Z}_a$ back to the dimensions of $\mathbf{X}_a$. For having a decoding scheme analogous to the encoding one, we employ another set of $N_{CNN}$ convolutional blocks for $d_a$, again with BN and ReLU, and using the same serial processing described by Eq. (3). This processing yields the learned feature maps of the decoder, $\mathbf{H}^{l_{da}} \in \mathbb{R}_{\geq 0}^{K_{l_{da}} \times T'_{l_{da}} \times F'_{l_{da}}}$, with $l_{da} = 1 + N_{CNN}, \ldots, 2N_{CNN}$ and $\mathbf{H}^{2N_{CNN}} = \hat{\mathbf{X}}_a$. To optimize $e_a$ and $d_a$, we employ the generalized KL divergence, $D_{KL}$, and we utilize the following loss function

$$\mathcal{L}_a(\mathbf{X}_a, \theta_{ea}, \theta_{da}) = D_{KL}(\mathbf{X}_a || \hat{\mathbf{X}}_a). \qquad (4)$$

Each audio signal represented by $\mathbf{X}_a$ is annotated by a set of tags from a vocabulary of size $C$. We want to exploit the semantics of each tag and, at the same time, capture the semantic relationships between tags. For that reason, we opt to use another AE structure, which outputs a latent learned representation of the set of tags of $\mathbf{X}_a$ as the learned features from the tags, and then tries to reconstruct the tags from that latent representation. Similar approaches have been used in (Silberer & Lapata, 2014), where an AE structure was employed in order to learn an embedding from a $k$-hot encoding of tags/words that would encapsulate semantic information. Specifically, we represent the set of tags for $\mathbf{X}_a$ as a multi-hot vector, $\mathbf{y}_t \in \{0, 1\}^C$. We use again an encoder $e_t$ and a decoder $d_t$, to obtain a learned latent representation of $\mathbf{y}_t$ as

$$\mathbf{z}_t = e_t(\mathbf{y}_t; \theta_{et}), \text{ and} \qquad (5)$$
$$\hat{\mathbf{y}}_t = d_t(\mathbf{z}_t; \theta_{dt}), \qquad (6)$$

where $\mathbf{z}_t \in \mathbb{R}_{\geq 0}^M$ is the learned latent representation of the tags for $\mathbf{X}_a$, $\mathbf{y}_t$ and $\hat{\mathbf{y}}_t$ is the reconstructed multi-hot encoding of the same tags $\mathbf{y}_t$. The $e_t$ consists of a set of trainable feed-forward linear layers, where each layer is followed by a BN and a ReLU, similar to Eq. 3. That is, if $\mathrm{FNN}^{l_t}$ is the $l_t$-th feed-forward linear layer, then

$$\mathbf{h}^{l_t} = \mathrm{ReLU}(\mathrm{BN}^{l_t}(\mathrm{FNN}^{l_t}(\mathbf{h}^{l_t-1}))), \qquad (7)$$

where $l_t = 1, \ldots, N_{FNN}$, $\mathbf{h}^{N_{FNN}} = \mathbf{z}_t$, and $\mathbf{h}^0 = \mathbf{y}_t$. To obtain the reconstructed version of $\mathbf{y}_t$, $\hat{\mathbf{y}}_t$, through $\mathbf{z}_t$, we use the decoder $d_t$, which is modeled analogously to $e_t$ and containing another set of $N_{FNN}$ feed-forward linear layers. $d_t$ processes $\mathbf{z}_t$ similarly to Eq. 7, with $\mathbf{h}^{1+N_{FNN}}$ to be the output of the first feed-forward linear layer of $d_t$, and $\mathbf{h}^{2N_{FNN}} = \hat{\mathbf{y}}_t$. To optimize $e_t$ and $d_t$ we utilize the loss $\mathcal{L}_t(\mathbf{y}_t, \theta_{et}, \theta_{dt}) = CE(\mathbf{y}_t, \hat{\mathbf{y}}_t)$, where $CE$ is the cross-entropy function.

## 2.2. Alignment of acoustic and semantic features

One of the main targets of our method is to infuse semantic information from the latent representation of tags to the learned acoustic features of audio. To do this, we maximize the agreement between (i.e. align) the paired latent representations of the audio signal, $\mathbf{Z}_a^q$, and the corresponding tags, $\mathbf{z}_t^q$, inspired by previous and relative work on image processing (Feng et al., 2014; Schonfeld et al., 2019), and by using a contrastive loss, similarly to (Sohn, 2016; Chen et al., 2020). Aligning these two latent representations (by pushing $\mathbf{Z}_a^q$ towards $\mathbf{z}_t^q$), will infuse $\mathbf{Z}_a^q$ with information from $\mathbf{z}_t^q$. This task is expected to be difficult, due to the fact that some acoustic aspects may not be covered by the tags, or that some existing tags may be wrong or not informative. Therefore, we utilize two affine transforms, and we align the outputs of these transforms. Specifically, we utilize the affine transforms $\mathrm{AFF}_a$ and $\mathrm{AFF}_t$, parameterized by $\theta_{af\text{-}a}$ and $\theta_{af\text{-}t}$ respectively, as

$$\boldsymbol{\Phi}_a = \mathrm{AFF}_a(\mathbf{Z}_a; \theta_{af\text{-}a}), \text{ and} \qquad (8)$$
$$\boldsymbol{\phi}_t = \mathrm{AFF}_t(\mathbf{z}_t; \theta_{af\text{-}t}). \qquad (9)$$

where $\boldsymbol{\Phi}_a \in \mathbb{R}_{\geq 0}^{K \times T' \times F}$ and $\boldsymbol{\phi}_t \in \mathbb{R}_{\geq 0}^M$. Then, since $\boldsymbol{\Phi}_a$ is a matrix and $\boldsymbol{\phi}_t$ a vector, we flatten $\boldsymbol{\Phi}_a$ to $\boldsymbol{\phi}_a \in \mathbb{R}_{\geq 0}^{KT'F'}$. To align $\boldsymbol{\phi}_a$ with its paired $\boldsymbol{\phi}_t$, we utilize randomly (and without repetition) sampled minibatches $\mathbb{G}_b = \{(\mathbf{X}_a^b, \mathbf{y}_t^b)\}_{b=1}^{N_b}$ from our dataset $\mathbb{G}$, where $N_b$ is the amount of paired examples in the minibatch $\mathbb{G}_b$. For each minibatch $\mathbb{G}_b$, we align the $\boldsymbol{\phi}_a^b$ with its paired $\boldsymbol{\phi}_t^b$ and, at the same time, we optimize $e_a$, $d_a$, $e_t$, $d_t$, $\mathrm{AFF}_a$ and $\mathrm{AFF}_t$. To do this, we follow (Chen et al., 2020) and we use the contrastive loss function

$$\mathcal{L}_\xi(\mathbb{G}_b, \boldsymbol{\Theta}_c) = \sum_{b=1}^{N_B} - \log \frac{\Xi(\boldsymbol{\phi}_a^b, \boldsymbol{\phi}_t^b, \tau)}{\sum_{i=1}^{N_b} \mathbb{1}_{[i \neq b]} \Xi(\boldsymbol{\phi}_a^b, \boldsymbol{\phi}_t^i, \tau)}, \text{ where} \qquad (10)$$

$$\Xi(\mathbf{a}, \mathbf{b}, \tau) = \exp(\mathrm{sim}(\mathbf{a}, \mathbf{b})\tau^{-1}), \qquad (11)$$
$$\mathrm{sim}(\mathbf{a}, \mathbf{b}) = \mathbf{a}^\top \mathbf{b} (||\mathbf{a}|| \, ||\mathbf{b}||)^{-1}, \qquad (12)$$

$\boldsymbol{\Theta}_c = \{\theta_{ea}, \theta_{af\text{-}a}, \theta_{et}, \theta_{af\text{-}t}\}$, $\mathbb{1}_A$ is the indicator function with $\mathbb{1}_A = 1$ iff A else 0, and $\tau$ is a temperature hyper-parameter. Finally, we jointly optimize $\theta_{ea}$, $\theta_{da}$, $\theta_{et}$, and $\theta_{dt}$, for each minibatch $\mathbb{G}_b$, minimizing

$$\mathcal{L}_{\mathrm{total}}(\mathbb{G}_b, \boldsymbol{\Theta}) = \lambda_a \sum_{b=1}^{N_B} \mathcal{L}_a(\mathbf{X}_a^b, \boldsymbol{\Theta}_a) + \lambda_t \sum_{b=1}^{N_B} \mathcal{L}_t(\mathbf{y}_t^b, \boldsymbol{\Theta}_t)$$
$$+ \lambda_\xi \mathcal{L}_\xi(\mathbb{G}_b, \boldsymbol{\Theta}_c), \qquad (13)$$

where $\boldsymbol{\Theta}_a = \{\theta_{ea}, \theta_{da}\}$, $\boldsymbol{\Theta}_t = \{\theta_{et}, \theta_{dt}\}$, $\boldsymbol{\Theta}$ is the union of the $\boldsymbol{\Theta}_\star$ sets in Eq. (13), and $\lambda_\star$ is a hyper-parameter used for numerical balancing of the different learning signals/losses. After the minimization of $\mathcal{L}_{\mathrm{total}}$, we use $e_a$ as a pre-learned feature extractor for different audio classification tasks.

## 3. Evaluation

We conduct an ablation study where we compare different methods for learning audio embeddings on their classifi-

cation performance at different tasks, using as input the embeddings from the employed methods. This allows us to evaluate the benefit of using the alignment and the reconstruction objectives in our method. We consider a traditional set of hand-crafted features, as a low anchor. Additionally, we perform a correlation analysis with a set of acoustic features in order to understand what kind of acoustic properties are reflected in the learnt embeddings.

### 3.1. Pre-training dataset and data pre-processing

For creating our pre-training dataset $\mathbb{G}$, we collect all sounds from Freesound (Font et al., 2013), that have a duration of maximum 10 seconds. We remove sounds that are used in any datasets of our downstream tasks. We apply a uniform sampling rate of 22 kHz and length of 10 secs to all collected sounds, by resampling and zero-padding as needed. We extract $F = 96$ log-scaled mel-band energies using sliding windows of 1024 samples ($\approx$46 ms), with 50% overlap and the Hamming windowing function. We create overlapping patches of $T = 96$ feature vectors ($\approx$2.2 s), using a step of 12 vectors for overlap. Then, we select the $T \times F$ patch with the maximum energy. This process is simple but we assume that in many cases, the associated tags will refer to salient events present in regions of high energy. We process the tags associated to the audio clips, by firstly removing any stopwords and making any plural forms of nouns to singular. We remove tags that occur in more than 70% of the sounds as they can be considered less informative, and consider the $C$=1000 remaining most occurring tags, which we encode using the multi-hot scheme. Finally, we discard sounds that were left with no tag after this filtering process. This process generated $Q = $189 896 spectrogram patches for our dataset $\mathbb{G}$. 10% of these patches are kept for validation and all the patches are scaled to values between 0 and 1.

We consider three different cases for evaluating the benefit of the alignment and the reconstruction objectives. The first is the method presented in Section 2, termed as AE-C. At the second, termed as E-C, we do not employ $d_a$ and $d_t$, and we optimize $e_a$ using only $\mathcal{L}_\xi$, similar to (Chen et al., 2020). The third, termed as CNN, is composed of $e_a$, followed by two fully connected layers and is optimized for directly predicting the tag vector $\mathbf{y}_t$ using the $CE$ function. Finally, we employ the 20 first mel-frequency cepstral coefficients (MFCCs) with their $\Delta$s and $\Delta\Delta$s as a low anchor, using means and standard deviations through time, and we term this case as MFCCs.

### 3.2. Downstream classification tasks

We consider three different audio classification tasks: i) sound event recognition/tagging (SER), ii) music genre classification (MGC), and iii) musical instrument classification (MIC). For SER, we use the Urban Sound 8K dataset (US8K) (Salamon et al., 2014) in our experiment, which consists of around 8000 single-labeled sounds of maximum

4 seconds and 10 classes. We use the provided folds for cross-validation. For MGC, we use the fault-filtered version of the GTZAN dataset (Tzanetakis & Cook, 2002; Kereliuk et al., 2015) consisting of single-labeled music excepts of 30 seconds, split in pre-computed sets of 443 songs for training and 290 for testing. Finally, for MIC, we use the NSynth dataset (Engel et al., 2017) which consists of more than 300k sound samples organised in 10 instrument families. However, because we are interested to see how our models performs with relatively low amount of training data, we randomly sample from NSynth a balanced set of 20k samples from the training set which correspond to approximately 7% of the original set. The evaluation set is kept the same.

For the above tasks and datasets, we use non-overlapping frames of audio clips that are calculated similarly to the pre-training dataset, and are given as input to the different methods in order to obtain the embeddings. Then, these embeddings are aggregated into a single vector (e.g. of 1152 dimensionality for our $e_a$) employing the mean statistic, and are used as an input to a classifier that is optimized for each corresponding task. Embeddings and MFCCs vectors are standardized to zero-mean and unit-variance, using statistics calculated from the training split of each task. As a classifier for each of the different tasks, we use a multi-layer perceptron (MLP) with one hidden layer of 256 features, similar to what is used in (Cramer et al., 2019). To obtain an unbiased evaluation of our method, we repeat the training procedure of the MLP in each task 10 times, average and report the mean accuracies.

### 3.3. Correlation analysis with acoustic features

We perform a correlation analysis using a similarity measure involving the Canonical Correlation Analysis (CCA) (Hardoon et al., 2004), to investigate the correlation of the output embeddings from our method, with various low-level acoustic features. Similar to (Raghu et al., 2017), we use sounds from the validation set of the pre-training dataset $\mathbb{G}$, and we compute the canonical correlation similarity (CCS) of our audio embedding $\mathbf{Z}_a$ with statistics of acoustic features computed with the librosa library (McFee et al., 2015). These features correspond to MFCCs, chromagram, spectral centroid, and spectral bandwidth, all computed at a frame level.

## 4. Results

In Table 1 are the results of the performance of the different embeddings and our MFCCs baseline, and results reported in the literature which are briefly explained in the supplementary material section. In all the tasks, AE-C and E-C embeddings yielded better results than the MFCCs baseline, showing that it is possible to learn meaningful audio representations, by taking advantage of tag metadata. However, the CNN case does not even reach the performance of the MFCCs features. This clearly indicates the benefit of

*Table 1.* Average mean accuracies for SER, MGC, and MIC. Additional performances are taken from the literature (Cramer et al., 2019; Salamon & Bello, 2017; Pons & Serra, 2019b; Lee et al., 2018; Ramires & Serra, 2019).

|          | US8K | GTZAN | NSynth |
|----------|------|-------|--------|
| MFCCs    | 65.8 | 49.8  | 62.6   |
| AE-C     | **72.7** | **60.7** | **73.1** |
| E-C      | 72.5 | 58.9  | 69.5   |
| CNN      | 48.4 | 47.0  | 56.4   |
| OpenL3   | 78.2 | –     | –      |
| VGGish   | 73.4 | –     | –      |
| DeepConv | **79.0** | –  | –      |
| rVGG     | 70.7 | 59.7  | –      |
| sampleCNN | –   | **82.1** | –   |
| smallCNN | –    | –     | **73.8** |

*Table 2.* CCA correlation scores between the embeddings model outputs and some acoustic features statistics.

|      | mean | var | skew | mean | var | skew |
|------|------|-----|------|------|-----|------|
|      | \multicolumn MFCCs | | | Chromagram | | |
| AE-C | **0.84** | **0.51** | **0.42** | 0.48 | **0.37** | 0.40 |
| E-C  | 0.58 | 0.49 | 0.39 | 0.38 | 0.36 | 0.32 |
| CNN  | 0.73 | 0.43 | 0.32 | **0.59** | 0.33 | **0.48** |
|      | Spectral Centroid | | | Spectral Bandwidth | | |
| AE-C | **0.97** | **0.87** | **0.80** | **0.96** | **0.86** | **0.84** |
| E-C  | 0.93 | 0.82 | 0.76 | 0.92 | 0.82 | 0.81 |
| CNN  | 0.95 | 0.76 | 0.74 | 0.91 | 0.72 | 0.80 |

our approach for building general audio representations by leveraging user-provided noisy tags. When comparing the different proposed embeddings, we see that the AE-C case consistently leads to better results. For the MIC (NSynth) task, combining reconstruction and contrastive objectives (i.e. AE-C case) brings important benefits. For the MGC (GTZAN) task, these benefits are not as pronounced, and finally, when looking at the SER (US8K) task, adding the reconstruction objective does not improve the results much. Our assumption is that recognizing musical instruments can be more easily done using lower-level features reflecting acoustic characteristics of the sounds, and that the reconstruction objective imposed by the autoencoder architecture is forcing the embedding to reflect low-level characteristics present in the spectrogram. However, for recognizing urban sounds or musical genres, a feature that reflects mainly semantic information is needed, which seems to be learned successfully when considering the contrastive objective.

Comparing our method to others for the SER, we can see that we are slightly outperformed by VGGish (Hershey et al., 2017; Gemmeke et al., 2017), according to results taken from (Cramer et al., 2019), which has been trained with million of manually annotated audio files using predefined categories. This shows that our approach which only takes advantage of small-scale content with their original tag metadata is very promising for learning competitive audio features. However, our model is still far from reaching performances given by OpenL3 or the current SOTA Deep-Conv with data augmentation. Similarly in MGC, the sam-

pleCNN classifier, pre-trained on the Million Song Dataset (MSD) (Lee et al., 2018) produces much better results than our approach. But, all these models have been either trained with much more data than ours, or use a more powerful classifier. Finally, NSynth dataset has been originally released in order to train generative models rather than classifiers. Still, results from (Ramires & Serra, 2019), show that our approach training using around 7% of the training data, is only slightly outperformed by a CNN trained with all the training data (smallCNN).

Table 2 shows the correlation for the different embeddings $Z_a$ with the mean, the variance, and the skewness of the different acoustic feature vectors. Overall, we observe a consistent increase of the correlation between the acoustic features and embeddings trained with models containing an AE structure. This suggests that the reconstruction objective enables to learn features that reflect some low-level acoustic characteristics of audio signals, which makes it more valuable as a general-purpose feature. More specifically, there is a large correlation increase between the mean of MFCCs and models that contain AE structure, showing that they can capture more timbral characteristics of the signal. However, variance and skwewness did not increase considerably, which can mean that our embeddings lack to capture temporal queues. Considering chromagrams, which reflect the harmonic contents of a sound, we see little improvement with AE models. This suggests that our embeddings lack some important musical characteristics. Regarding the spectral centroid and bandwidth, we only observe a slight increase of correlations with AE-based embeddings.

## 5. Conclusions

In this work we present a method for learning an audio representation that can capture acoustic and semantic characteristics for a wide range of sounds. We utilise two heterogeneous autoencoders (AEs), one taking as an input audio spectrogram and the other processing a tag representation. These AEs are jointly trained and a contrastive loss enables to align their latent representations by leveraging associated pairs of audio and tags. We evaluate our method by conducting an ablation study, where we compare different methods for learning audio representations over three different classification tasks. We also perform a correlation analysis with acoustic features in order to grasp knowledge about what type of acoustic characteristics the embedding captures.

Results indicate that combining reconstruction objectives with a contrastive learning framework enables to learn audio features that reflect both semantic and lower-level acoustic characteristics of sounds, which makes it suitable for general audio machine listening applications. Future work may focus on improving the network models by for instance using audio architectures that can capture more temporal aspects and dynamics present in audio signals.

# Supplementary Material

## Code and data

The code of our method is available online at: `https://github.com/xavierfav/coala`. We provide the pre-training dataset $\mathbb{G}$ online and publicly at: `https://zenodo.org/record/3887261`. Sounds were accessed from the Freesound API on the 7th of May, 2019.

## Utilized hyper-parameters, training procedure, and models

For the audio autoencoder, we use $N_{\text{CNN}}$=5 convolutional blocks each one containing $K_{l_{ea}} = 128$ filters of shape 4x4, with a stride of 2x2, yielding an embedding $\phi_a$ of size 1152. This audio encoder model has approximately 2.4M parameters. The tag autoencoder is composed of $N_{\text{FNN}}$=3 layers of size 512, 512 and 1152, accepting a multi-hot vector of dimension 1000 as input. We train the models for 200 epochs using a minibatch size $N_{\text{B}}$=128, using an SGD optimizer with a learning rate value of 0.005. We utilize the validation set to define the different $\lambda$'s at Eq. (13) and the constrastive loss temperature parameter $\tau$, to $\lambda_a=\lambda_t$=5, $\lambda_\xi$=10, and $\tau = 0.1$. We add a dropout regularization with rate 25% after each activation layer to avoid overfitting while training. The CNN baseline that is trained by predicting directly the multi-hot tag vectors from the audio spectrogram has follows the same architecture as the encoder from the audio autoencoder. When training, we add 2 fully connected layers and train it for 20 epochs using a minibatch size $N_{\text{B}}$=128 and an SGD optimizer with a learning rate value of 0.005 as well.

## Tag processing

Removing stop-words in sound tags is done using the NLTK python library (`https://www.nltk.org/`). Making any plural forms of nouns to singular is done with the inflect python library (`https://github.com/jazzband/inflect`). Additionally we transform all tags to lower-case.

## Models from the literature

OpenL3 (Cramer et al., 2019) is an open source implementation of Look, Listen, and Learn (L3-Net) (Arandjelovic & Zisserman, 2017). It consists of an embedding model using blocks of convolutional and max-pooling layers, trained through self-supervised learning of audio-visual correspondence in videos from YouTube. The model has around 4.7M parameters and computes embedding vectors of size 6144. In (Cramer et al., 2019), the authors report the classification accuracies of different variants of the model used as a feature extractor combined with a MLP classifier on the US8K dataset. Their mean accuracy is 78.2%.

VGGish (Hershey et al., 2017; Gemmeke et al., 2017) consists of an audio-based CNN model, a modified version of the VGGNet model (Simonyan & Zisserman, 2014) trained to predict video tags from the Youtube-8M dataset (Abu-El-Haija et al., 2016). The model has around 62M parameters and computes embedding vectors of size 128. Its accuracy when used as a feature extractor combined with a MLP classifier on the US8K dataset is reported in (Cramer et al., 2019) as being 73.4%.

DeepConv (Salamon & Bello, 2017) is a deep neural network composed of convolutional and max-pooling layers. When trained with data augmentation on the US8K dataset, it achieved 79.0% accuracy.

rVGG (Pons & Serra, 2019b) corresponds to a VGGish non-trained model (randomly weighted). The referenced work experiment using it as a feature extractor by comparing different embeddings from different layers of the network. The best accuracies on US8K and GTZAN (fault-filtered) when combined with an SVM classifier were reported as 70.7% and 59.7% respectively, using an embedding vector of size of 3585.

sampleCNN (Lee et al., 2018) is a deep neural network that takes as input the raw waveform and is composed of many small 1D convolutional layers and that has been designed for musical classification tasks. When pre-trained on the Million Song Dataset (Bertin-Mahieux et al., 2011), this model reached a 82.1% accuracy on the GTZAN dataset (fault-filtered).

smallCNN (Pons et al., 2017b) is a neural network composed of one CNN layer with filters of different sizes that can capture timbral characteristics of the sounds. It is combined with pooling operations and a fully-connected layer in order to predict labels. In (Ramires & Serra, 2019), it has been trained with the NSynth dataset in order to predict the instrument family classes and was reported to reach 73.8% accuracy.

## Acknowledgement

X. Favory, K. Drossos, and T. Virtanen would like to acknowledge CSC Finland for computational resources. The authors would also like to thank all the Freesound users that have been sharing very valuable content for many years. Xavier Favory is also grateful for the GPU donated by NVidia.

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
