# OpenReview forum: "COALA: Co-Aligned Autoencoders for Learning Semantically Enriched Audio Representations"
_ICML.cc/2020/Workshop/SAS — SAS 2020_

### Official Review · AnonReviewer1 · 2020-06-29
**Some novelty in the idea, but not SOTA results reported**

**Rating:** 5
**Confidence:** 4

**Review:**

Interesting method, but results still far from SOTA. Also, interesting ablation study to see the influence of each element to the embeddings’ performance

It seems that you have some data with tags, some without, right? What percentage of your data should be with tags for the proposed method to work?

Table 1: better results when using tags, but still far from state-of-the-art. What is then the benefit of the proposed approach? Compared to VGGish, less data and original tag metadata explains the fall in performance.
About the SOTA methods you state that “all these models have been either trained with much more data than ours, or use a more powerful classifier. “ -> How could the gap been closed with SOTA approaches?
Use pretrain model since you have less data? Do some cleaning of noisy tags? Use a more powerful classifier in your approach?

“This suggests that our embeddings lack some important musical characteristics. “ -> Do you have any intuition why this is the case?

After reading the results section, I think the claim “Our results show that our method is in par with the state-of-the-art in the considered tasks” should be removed from the abstract.

Typos
Section 1: in order to co-regularizing -> in order to co-regularize
Section 3.2 To obtain an unbiased evaluataion…-> …evaluation…
Section 4: skwewnes -> skewness

---

### Official Review · AnonReviewer3 · 2020-06-29
**An approach to learning acoustic embeddings using autoencoders and a contrastive loss**

**Rating:** 6
**Confidence:** 4

**Review:**

**Pros**
+ Interesting framework for incorporating text information in learning acoustic embeddings
+ Analysis of the learned embeddings

**Cons**
- Approach seems only loosely related to the cited SimCLR approach
- Due to differences in data and architectures, it is difficult to compare to previous work

The paper describes an interesting framework for utilizing text information to improve the learning of embeddings for audio. The learned embeddings are further used for three downstream tasks. The paper is clearly written and the analysis gives some insight into the approach. I think the connection to SimCLR is not clear. Related to SimCLR, there are some ablation experiments that would have helped with the analysis. Due to the variety of model types and datasets used in previous work, it is difficult to understand how this approach compares to other work. Detailed comments are below.
It is difficult to judge the framework here with competing work in the literature. Table 1 contains results, but the supplementary material reveals that this systems are not directly comparable. The approaches all use different types of architectures and data. As an example, the proposed model performs worse than previous work on US8K. On one hand this work had the advantage of using the Freesound database for retraining, but on the other hand, it had the disadvantage of using a vastly simpler network.

Under contributions, the authors state "We adapt a recently introduced constrastive loss framework (Chen et al., 2020), and we apply it for audio representation learning in a heterogeneous setting". The work here differs from SimCLR in many respects. I would argue that if someone was asked to adapt SimCLR to audio, the framework would not look like this. There are some similarities though.

**Similarities**
- A temperature parameter is used in the objective function.
- A contrastive loss function.

**Differences**
- Use of linear transforms of the embedded features before the loss. The SimCLR paper uses a nonlinear network in that location. This is one of the features that set it apart from previous work.
- Use of a reconstruction loss. The SimCLR approach relies only on the alignment and does not contain a reconstruction loss.
- No augmentation. SimCLR utilizes data augmentation to learn an embedding that is robust to data transformations.
- Small batch size. The SimCLR paper is clear that larger batch sizes give significant gains. Here the batch size is 128, smaller than anything considered in the SimCLR paper. I understand this was likely a constraint imposed by available hardware.

There are many papers that take this general approach of using a contrastive loss. Other than the use of a temperature, I struggle to see what relates the work specifically to SimCLR as opposed to any other paper.

Based on the above comments, I think there are some additional ablations/alternatives that could have been performed that would have been illustrative.
- How does the model perform if the auto-encoder aspect is removed completely?
- How does performance change with a SimCLR-like setup, where the alignment is between two augmented versions of the audio input? How does this compare to the autoencoder?
- Does incorporating the text on top of this still help?
- How does varying the batch size impact performance?

I think the autoencoder approach, especially in conjunction with requiring agreement with the text embeddings is problematic. This requires the embedding based on the text to have enough information in order to reconstruct an acoustic signal. An acoustic signal contains a tremendous amount of information not captured by these tags. I understand the motivation for the autoencoder comes from previous work like Van Den Oord, 2017, but later work from that author "Representation Learning with
Contrastive Predictive Coding" argues against it.

---

### Official Review · AnonReviewer2 · 2020-06-29

**Rating:** 7
**Confidence:** 3

**Review:**

The authors propose a model for audio representation learning that uses 1) reconstructive learning, to train an audio encoder and a tag encoder as autoencoders, and 2) contrastive learning, to make the representation of the audio encoder and tag encoder match each other for paired data. This yields a model that outperforms MFCC features and can outperform a standard CNN even using only 7% of the data.

Using both types of learning seems like a great idea to me, and I'm surprised it hadn't been done before! I'd like to see comparisons with some other unsupervised models, like contrastive predictive coding, but I'd say this is a good start and definitely interesting enough for this workshop.

---

### Decision · Program_Chairs · 2020-07-01

**Decision:**

Accept

**Comment:**

Dear author(s),

Thank you very much for your submission at the ICML2020@SaS workshop (https://icml-sas.gitlab.io/). Based on the scores assigned by the reviewers, we are happy to notify you that your paper was accepted for the workshop.

Please, address the comments of the reviewers and submit the camera-ready version by July 8. We ask the authors to record a 15min video for your talk. At the workshop, we will have the pre-recorded video as well as a live QA session. It is important to keep this time limit, otherwise, your talk will be automatically cut. The deadline for uploading the video is July 8. The detailed instructions for uploading will follow.

Feel free to contact us for any questions!

Best,

The ICML20@SaS organizers:
Mirco Ravanelli
Titouan Parcollet
Dmitriy Serdyuk
Devon Hjelm
Bhuvana Ramabhadran